

# Predicting population: development and validation of a new predictive nomogram for evaluating medication nonadherence risk in a type 2 diabetes

NaRen QiMuge[1,2], Xu Fang[1], Baocheng Chang[1], Dong Mei Li[2] and Yuanyuan Li[2]

[1] NHC Key Laboratory of Hormones and Development, Tianjin Key Laboratory of Metabolic Diseases, Chu Hsien-I Memorial Hospital & Tianjin Institute of Endocrinology, Tianjin Medical University, Tianjin, Tianjin, China
[2] Department of Endocrinology and Metabolism, Inner Mongolia People's Hospital, Hohhot, Inner Mongolia Autonomous Region, China

Corresponding authors
Baocheng Chang,
826011502@qq.com
Dong Mei Li, ldm1229@126.com

## ABSTRACT

**Background:** Diabetes mellitus is a growing global health challenge and affects patients of all ages. Treatment aims to keep blood glucose levels close to normal and to prevent or delay complications. However, adherence to antidiabetic medicines is often unsatisfactory.

**Purpose:** Here, we established and internally validated a medication nonadherence risk nomogram for use in Chinese type 2 diabetes mellitus (T2DM) patients.

**Methods:** This cross-sectional study was carried out from July–December 2020 on randomly selected T2DM patients visiting a diabetes clinic and included 753 participants. Adherence was analyzed based on an eight-item Morisky Medication Adherence Scale (MMAS-8). Other data, including patient demographics, treatment, complications, and comorbidities, were also collected on questionnaires. Optimization of feature selection to develop the medication nonadherence risk model was achieved using the least absolute shrinkage and selection operator regression model (LASSO). A prediction model comprising features selected from LASSO model was designed by applying multivariable logistic regression analysis. The decision curve analysis, calibration plot, and $C$-index were utilized to assess the performance of the model in terms of discrimination, calibration, and clinical usefulness. Bootstrapping validation was applied for internal validation.

**Results:** The prediction nomogram comprised several factors including sex, marital status, education level, employment, distance, self-monitoringofbloodglucose, disease duration, and dosing frequency of daily hypoglycemics (pills, insulin, or glucagon-like peptide-1). The model exhibited good calibration and good discrimination ($C$-index = 0.79, 95% CI [0.75–0.83]). In the validation samples, a high $C$-index (0.75) was achieved. Results of the decision curve analysis revealed that the nonadherence nomogram could be applied in clinical practice in cases where the intervention is decided at a nonadherence possibility threshold of 12%.

**Conclusion:** The number of patients who adhere to anti-diabetes therapy was small. Being single male, having no formal education, employed, far from hospital, long disease duration, and taking antidiabetics twice or thrice daily, had significant

negative correlation with medication adherence. Thus, strategies for improving adherence are urgently needed.

# INTRODUCTION

In 2019, nearly 463 million cases of diabetes were documented worldwide, and about a quarter of these patients (129.8 million) were in China according to the International Diabetes Federation (IDF) (*Li et al., 2020*). While antidiabetics are vital in managing type 2 diabetes mellitus (T2DM), their effectiveness largely depends on the level of adherence to prescribed treatments.

Medication adherence rate (MAR) is defined as the degree to which patients follow prescribed regimens (*Osterberg & Blaschke, 2005*). A systematic review of patients with T2DM receiving oral hypoglycaemic drugs for 6–12 months found that MAR ranged from 36% to 93% (*Cramer, 2004*). Medication adherence is key to the optimal management of T2DM (*Bailey & Kodack, 2011*). Several factors, including disease, drug combinations, depression, stress, sex, age, and treatment features influence treatment adherence in patients with diabetes (*Krass, Schieback & Dhippayom, 2015*). Low medication adherence results in poor glycaemic control, and hence increases the risk of developing diabetic complications, such as nephropathy (*Fukuda & Mizobe, 2017*). It is crucial to design tools for predicting adherence rates and early interventions which may increase MAR. Thus, it is vital to enhance MAR, thereby achieving effective control of blood glucose and minimise diabetes complications.

A nomogram, a graphical tool, is designed to approximate quickly complicated calculations and is a viable tool for describing an individual's prognosis or risk of a clinical event (*Lasonos et al., 2008*). The combination of individualised therapy and widespread availability *via* the web has contributed to their popularity among clinicians and patients. Here, we used a nomogram to develop a simple prediction tool for assessing MAR. Although these features are present at the beginning of therapy, to the best of our knowledge, no studies have focused on this issue. In patients with diabetes, ensuring adequate medication intake based on monitoring tools such as the developed nomogram contributes to better diagnosis and prognosis as clinicians gain information on adherence patterns so that they can prevent complications and comorbidities.

# PATIENTS AND METHODS

## Patients

Participants were recruited from among 18–80-year-old individuals visiting the Inner Mongolia People's Hospital between July and December 2020. The included patients met the WHO (1999) diagnostic criteria for T2DM and had been on antihyperglycaemic agents. This study was approved by the local ethics committee of the Inner Mongolia People's Hospital (NO.202000406). All participants provided written informed consent,
filled questionnaires evaluating treatment adherence, and were involved in the 10-min questionnaire. The questionnaire consisted of items covering demographic characteristics (*e.g.*, sex, height, and body weight), disease characteristics (*e.g.*, diabetes duration, family history of diabetes), and treatment features (*e.g.*, the class of hypoglycaemic pills used daily, daily dosing frequency of hypoglycaemic agents). The doctors guided the participants in completing the questionnaire in the plain language. Illiterate patients were excluded, as were those with severe cognitive conditions or disabling physical constraints.

## Adherence assessment

The Morisky Medication Adherence Scale (MMAS)-based prediction nomogram may identify patients with diabetes with low medication adherence. MMAS is an established tool for assessing patients' medication adherence in an eight-question format (MMAS-8) by evaluating factors such as whether patients forget to take medication at home or when travelling, if they skip medication due to its unpleasant effects, if they stop medication because they feel healthy, or if they fail to take medication because it is inconvenient or difficult to remember (*Morisky, Green & Levine, 1986*). Responses to these items are dichotomous (yes/no), with the last item containing a five-point Likert response. We categorised adherence as follows: low adherence $\leq 6$, $6 \leq$ medium adherence $< 8$, and high adherence $\geq 8$. Patients with low or medium adherence were considered non-adherent.

## Statistical analyses

Demographics, treatment, and disease characteristics are presented as counts (%). Participants were randomly divided into a modelling and validation group at a 7:3 ratio using a computer program, and the modelling group was used to establish the nomogram, and the validation group was used for validation. Statistical analyses were performed using R software (Version 3.6.3).

The logistic least absolute shrinkage and selection operator (LASSO) model, a shrinkage method, can actively select from a large and potentially multicollinear set of variables in the regression, thereby obtaining a more relevant and interpretable set of predictors. The LASSO method was utilised to select optimal predictive risk factors from modelling patients with diabetes by reducing high-dimensional data (*Sauerbrei, Royston & Binder, 2007*; *Friedman, Hastie & Tibshirani, 2010*). LASSO is performed by successive shrinkage operations to minimise the regression coefficients to reduce the possibility of overfitting; however, the technique is calculated by shrinking the sum of the absolute values of the regression coefficients, forcing and producing coefficients to be exactly zero, thereby selecting non-zero variables to be retained in the model (*Kidd et al., 2018*). Next, multivariable logistic regression analysis was applied to construct the predictive model by incorporating the characteristics selected in the LASSO regression model. The characteristics were considered as odds ratios (ORs) with 95% confidence intervals (CIs). A two-sided test was used as the statistical significance tool. Sociodemographic variables (*p*-value $\leq 0.05$) associated with disease and treatment characteristics were included in the model (*Xing et al., 2017*). All potential predictors were applied to build predictive

models for the risk of medication nonadherence using the cohort (*Balachandran et al., 2015*; *Iasonos et al., 2008*).

Calibration curves were plotted to assess the calibration of the non-adherence nomogram. A significant test statistic implies that the model is not perfectly calibrated (*Kramer & Zimmerman, 2007*). Harrell's *C*-index was used to quantify the discrimination performance of the nonadherence nomogram. The non-adherence nomogram was tested on a validation sample to calculate a relatively corrected *C*-index (*Pencina & D'Agostino, 2004*). Decision curve analysis (DCA) was performed to determine the clinical utility of the nonadherence nomogram by quantifying the net benefits at different threshold probabilities in the diabetes cohort (*Vickers et al., 2008*). DCA is a simple method for evaluating clinical predictive models, diagnostic tests, and molecular markers. Its advantage is that it integrates the preferences of patients or decision makers into the analysis (*Fitzgerald, Saville & Lewis, 2015*). Doctors may be particularly concerned about missing disease after consultation and discussion with some patients. However, doctors may be more concerned about avoiding unnecessary intervention for other patients. Doctors may also differ in their propensity to intervene, more conservative, or more aggressive (*Vickers, Calster & Steyerberg, 2019*). DCA is a method for assessing and comparing prediction models based on clinical consequences. It is based on the principle that the probability of a doctor recommending treatment is informed by how doctors and patients weigh the harm of a false-positive outcome against the harm of a false-negative outcome. Investigators should first consider the relative harm of avoiding interventions for patients with disease and unnecessary interventions for patients without disease in order to develop a clinically reasonable range of threshold probabilities. Then, they should decide whether the net benefit of their model or test is more appropriate than the alternatives within this threshold probability range. In a similar way, decision curves are applied to assess whether a model or test would be beneficial in clinical practice. If the results are positive, the model or test can be applied to appropriate patients as part of a shared decision (*Huang et al., 2016*).

## RESULTS

### Characteristics of patients

A total of 753 patients (438 men and 315 women; mean age: 57.12 ± 12.25 years, range 20–79 years) who visited our clinic between July and December 2020 completed the questionnaire. Based on the MMAS-8 score, patients were divided into two groups: adherence and non-adherence groups. Patient data, including demographics, disease characteristics, and treatment features, in the two groups are shown in Table 1.

### Feature selection

Data regarding demographics (age, sex, BMI, marital status, education level, employment, working strength, monthly *per capita* income, type of medical insurance, distance to hospital), disease characteristics (disease duration, family history of diabetes, complications, comorbidities), treatment features (types of all prescribed daily, types of

**Table 1 Differences between demographic and clinical characteristics of adherent and nonadherent groups.**

| Demographic characteristics | n (%) | | |
| --- | --- | --- | --- |
| | Adherence (n = 374) | Nonadherence (n = 379) | Total (n = 753) |
| Age (years) | | | |
| 20–39 | 35 | 39 | 74 |
| 40–59 | 172 | 189 | 361 |
| 60–79 | 167 | 151 | 318 |
| Sex | | | |
| male | 197 | 241 | 438 |
| Female | 177 | 138 | 315 |
| BMI (Kg/m$^2$) | | | |
| <18.5 | 22 | 16 | 38 |
| 18.5~<24.0 | 127 | 123 | 250 |
| 24.0~<28.0 | 145 | 164 | 309 |
| ≥28.0 | 80 | 76 | 156 |
| Marital status | | | |
| Married | 354 | 325 | 679 |
| Single | 20 | 54 | 74 |
| Education level | | | |
| Primary (0–9 years) | 144 | 165 | 309 |
| Secondary (9–12 years) | 213 | 207 | 420 |
| Higher (>12 years) | 17 | 7 | 24 |
| Employment | | | |
| Employed | 121 | 154 | 275 |
| Unemployed | 253 | 225 | 478 |
| Working strength | | | |
| Less activity (office, and so on) | 262 | 269 | 531 |
| Light-to-moderate activity (installer and so on) | 93 | 94 | 187 |
| moderate or heavy activity (agriculture and so on) | 19 | 16 | 35 |
| Monthly per capita income (yuan) | | | |
| <1,000 | 61 | 58 | 119 |
| 1,000–9,999 | 305 | 307 | 612 |
| 10,000–19,999 | 8 | 14 | 22 |
| Type of medical insurance | | | |
| Rural cooperative medical care | 271 | 278 | 549 |
| Urban medical insurance | 95 | 92 | 187 |
| Self-funded | 8 | 9 | 17 |
| Distance to hospital (Km) | | | |
| ≥30 | 55 | 61 | 116 |
| <30 | 319 | 318 | 637 |
| Self-monitoring of blood glucose (≥5 times/week) | | | |
| Yes | 333 | 307 | 640 |
| No | 41 | 72 | 113 |

(Continued)

| Table 1 (continued) | | | |
|---|---|---|---|
| Demographic characteristics | n (%) | | |
| | Adherence (n = 374) | Nonadherence (n = 379) | Total (n = 753) |
| **Disease characteristics** | | | |
| Disease duration (years) | | | |
| <1 | 75 | 44 | 119 |
| 1–5 | 60 | 74 | 134 |
| 5–10 | 70 | 113 | 183 |
| ≥10 | 169 | 148 | 317 |
| Family history of diabetes | | | |
| Yes | 153 | 161 | 314 |
| No | 221 | 218 | 439 |
| Complications | | | |
| Yes | 63 | 65 | 128 |
| No | 221 | 218 | 439 |
| Unknown | 90 | 96 | 186 |
| Comorbidities | | | |
| Yes | 90 | 80 | 170 |
| No | 230 | 251 | 481 |
| Unknown | 54 | 48 | 102 |
| **Treatment features** | | | |
| Types of pills prescribed daily | | | |
| 1–2 | 213 | 212 | 425 |
| 3 | 62 | 64 | 126 |
| 4–5 | 56 | 59 | 115 |
| ≥6 | 43 | 44 | 87 |
| Types of hypoglycemic pills prescribed daily | | | |
| 0 | 104 | 116 | 220 |
| 1 | 109 | 117 | 226 |
| 2 | 87 | 89 | 176 |
| ≥3 | 74 | 57 | 131 |
| Current use of insulin or/and GLP-1 | | | |
| Yes | 294 | 296 | 590 |
| No | 80 | 83 | 163 |
| Dosing frequency daily of hypoglycemic agents (pills, insulin, GLP-1) | | | |
| Once | 116 | 42 | 158 |
| Twice | 47 | 66 | 113 |
| Thrice | 47 | 146 | 193 |
| ≥quartic | 165 | 125 | 290 |

**Note:**
  GLP-1, glucagon-like peptide 1.

hypoglycaemic pills prescribed daily, current use of insulin and/or GLP-1, and frequent dosing of hypoglycaemic agents) were considered potential prognostic factors affecting medication adherence and were included in the LASSO regression. Regarding

demographics, disease, and treatment features, 19 features were reduced to eight potential predictors since the 753 patients (Figs. 1A and 1B) had nonzero coefficients based on LASSO regression analysis. These features included sex, marital status, education level, employment, distance, self-monitoring of blood glucose (SMBG), disease duration, and daily dose frequency of hypoglycaemic agents (pills, insulin, or glucagon-like peptide 1 [GLP-1]).

## Development of an individualized prediction model

Logistic regression analysis results of sex, marital status, education level, employment, distance, SMBG, disease duration, and daily dosing frequency of hypoglycaemic agents (pills, insulin, or GLP-1) are shown in Table 2. The risk of medication nonadherence was negatively related with being a woman ($\beta = -0.62$, OR = 0.54, 95% CI [0.14–0.90]), secondary ($\beta = -0.53$, OR = 0.59, 95% CI [0.37–0.93]) and higher ($\beta = -1.47$, OR = 0.23, 95% CI [0.06–0.72]) education, unemployment ($\beta = -0.37$, OR = 0.69, 95% CI [0.43–1.12]), distance to hospital (<30 km) ($\beta = -0.56$, OR = 0.57, 95% CI [0.32–1.01]). The risk of medication nonadherence was positively correlated with single ($\beta = 0.62$, OR = 0.54, 95% CI [0.14–0.90]), self-monitoring of blood glucose times ≤5/week ($\beta = 1.17$, OR = 3.23, 95% CI [1.81–5.97]), disease duration of 5–10 years ($\beta = 1.57$, OR = 4.8, 95% CI [2.42–9.85]), and three times daily of hypoglycaemic agents (pills, insulin, GLP-1) ($\beta = 2.15$, OR = 8.59, 95% CI [4.68–16.21]). However, some risk factors' confidence intervals are wide because of the small sample sizes; meanwhile, the magnitude of the confidence interval reflects the contribution to the outcome event. Next, a model based on these independent predictors was developed (Fig. 2). The nomogram was developed by graphically assigning an initial score to each of the eight independent prognostic factors, ranging from 0 to 100. All scores were drawn as a vertical line from the total points, indicating the estimated probability of medication nonadherence. A higher total score in the nomogram was predicted to be associated with a higher probability of medication nonadherence, whereas a lower total score was associated with a lower probability of medication nonadherence.

## Apparent performance of the nonadherence risk nomogram in the cohort

The calibration curve of the risk nomogram for predicting T2DM medication nonadherence revealed a decent agreement in this cohort (Fig. 3). The C-index of the prediction nomogram was 0.79 (95% CI [0.75–0.83]) for the modelling cohort, and 0.75 in the validation samples, indicating the good discrimination of this model. Apparent performance exhibited good prediction capability in the non-adherence risk nomogram.

## Clinical use

DCA of the medication nonadherence nomogram revealed that if the threshold probability was >12% and <85%, using this nonadherence nomogram to predict medication nonadherence risk and take intervention adds more benefit than intervening all or none of the patients (Fig. 4). This shows that the nomogram has a good clinical value.
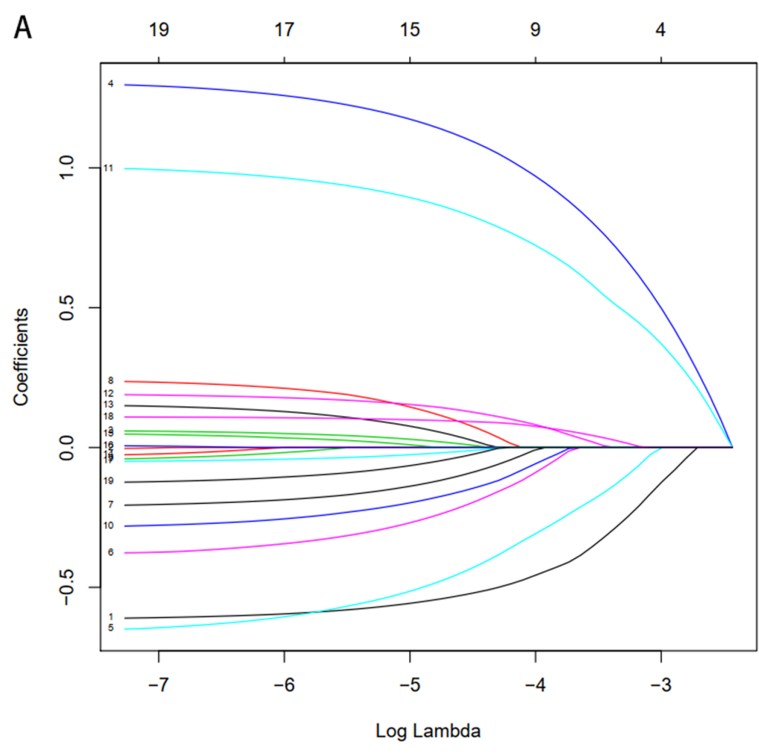

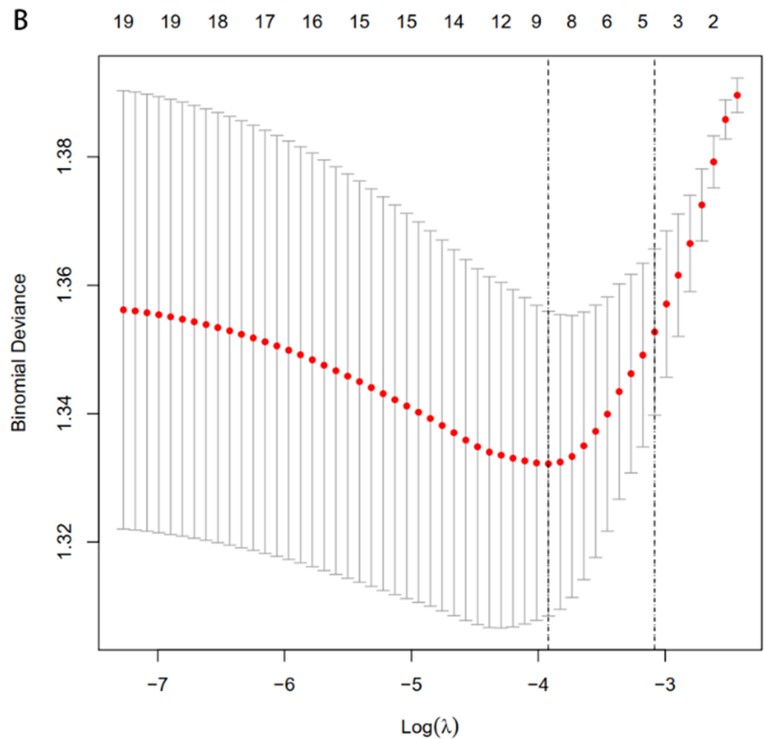

**Figure 1 Demographic and clinical feature selection using the LASSO binary logistic regression model.** (A, B) Results on the 19 variables included in the LASSO regression and their corresponding coefficients for the different values of the variable. (A) The model coefficient trendlines of the 19 features. Each line in graphs represents one variable, the vertical axisis the estimated coefficient of the variables, ordinate displays tuning parameter log (lambda) sequence. Different lambda values had different candidate

**Figure 1** (continued)
variables. Optimal parameter (lambda) selection in the LASSO model used fivefold cross-validation *via* minimum criteria (10,17). Specific correlation coefficient for each measured variable was determined using coef (cvfit, s = lambda). Variables with non zero coefficients are the result of the screening. (B) Depiction of the process of selecting optimal parameters by LASSO regression. The abscissa represents logarithm of parameter λ and the ordinate represents the model errors, the numbers on the top of the figures indicated the number of the candidate variables for the corresponding lambda (λ) value in LASSO regression. The number of variable varied according to the estimated value of λ. Abbreviations: LASSO, least absolute shrinkage and selection operator; SE, standard error.

**Table 2 Prediction factors for medication nonadherence in type 2 diabetes patients.**

| Intercept and variable | Prediction model | |
| --- | --- | --- |
| | β | Odds ratio (95% CI) |
| Intercept | −1.01 | 0.37 [0.14;0.90] |
| Sex: female (*vs* male) | −0.62 | 0.54 [0.3;0.85] |
| Marital status: Single (*vs* married) | 1.50 | 4.50 [2.21;9.74] |
| Education level (*vs* primary) | | |
| Secondary | −0.53 | 0.59 [0.37;0.93] |
| Higher | −1.47 | 0.23 [0.06;0.72] |
| employment: Unemployed (vs employed) | −0.37 | 0.69 [0.43;1.12] |
| Distance to hospital: <30 Km (*vs* ≥30 km ) | −0.56 | 0.57 [0.32;1.01] |
| Self-monitoring of blood glucose (≥5 times/week):No (*vs* Yes) | 1.17 | 3.23 [1.81;5.97] |
| Disease duration (years) (*vs* <1) | | |
| 1–5 | 1.02 | 2.76 [1.38;5.67] |
| 5–10 | 1.57 | 4.80 [2.42;9.85] |
| ≥10 | 0.89 | 2.43 [1.29;4.73] |
| Dosing frequency daily of hypoglycemic agents (pills, insulin, GLP-1) (*vs* Once) | | |
| Twice | 1.52 | 4.58 [2.37;9.08] |
| Thrice | 2.15 | 8.59 [4.68;16.21] |
| ≥Quartic | 0.67 | 1.96 [1.14;3.42] |

**Note:**
β is the regression coefficient.
CI, confidence interval.

The medication nonadherence model can then be utilised in appropriate patients as part of shared decision-making. For example, at the 20% risk cut-off and defining it as medication nonadherence, the net benefit was about 35%, which was equivalent to performing 35 out of every 100 cases benefiting from it without harming the interests of others.

## DISCUSSION

Similar to previous studies (*Krass, Schieback & Dhippayom, 2015*; *Wei et al., 2017*), almost 50% of the patients were nonadherent to their T2DM treatment regimens, which increases the risk of diabetes complications such as cardiovascular events, stroke, and chronic kidney

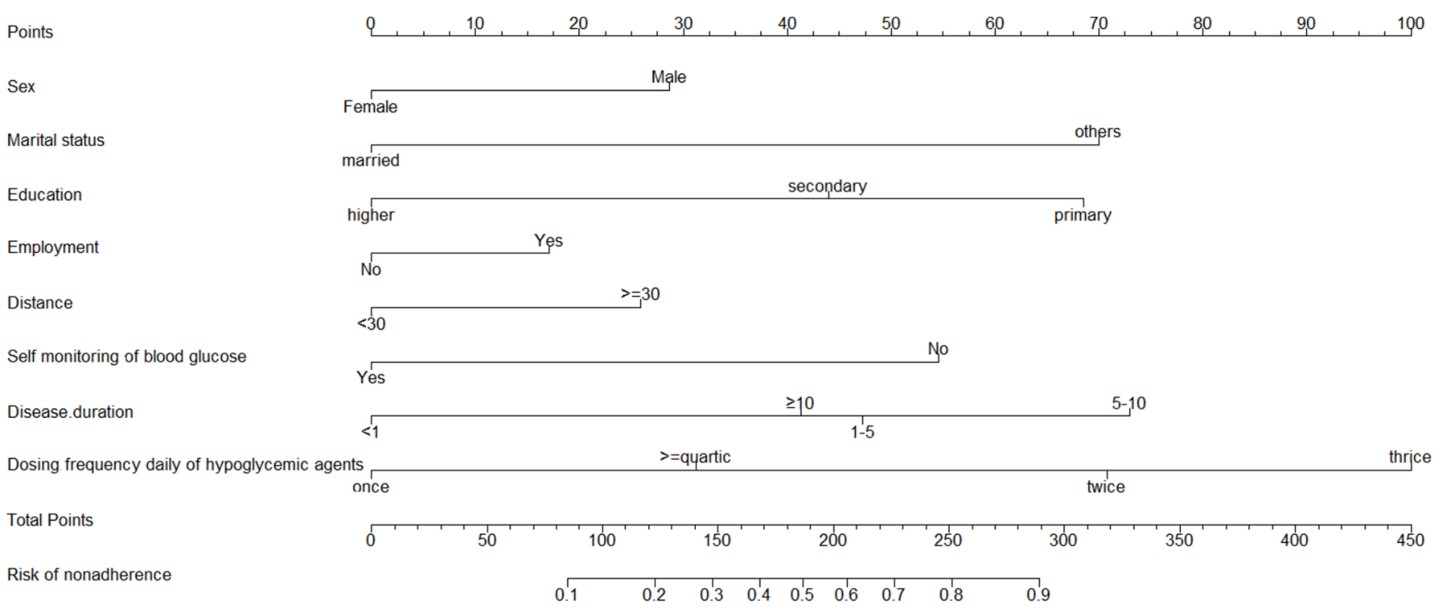

**Figure 2 Developed medication nonadherence nomogram.** The medication nonadherence nomogram was developed in the cohort, with the sex, marital status, education level, employment, distance, self-monitoring of blood glucose, disease duration, and dosing frequency daily of hypoglycemic agents (pills, insulin or GLP-1). The line segment corresponding to each variable is marked with a scale, which represents the possible value range of the variable, and the length of the line segment reflects the contribution of the variable to the outcome event. The value of each variable was gave a score on the point scale axis. A total score could be easily calculated by adding each single score and, by projecting the score to the lower total point scale, we were able to estimate the probability of medication nonadherence.

disease. Medication nonadherence is multifactorial and is affected by factors such as patients' beliefs, health literacy, socioeconomic status, and race/ethnicity (*Buckley, Labonville & Barr, 2016*; *Lewis, 2011*; *Geest et al., 2014*; *Conn et al., 2016*; *Pladevall et al., 2010*; *Ghembaza et al., 2014*). We found that 97% of the participants received anti-diabetes medication paid for by Medicare. Thus, unlike the study that reported that the most common reason for non-adherence to their medication was unavailable and unaffordable due to the local cost of antidiabetic medications (*Kassahun et al., 2016*).

The strength of the current study is the large number of patient-, disease-, and treatment-related variables that were evaluated. Patient-related variables included patient demographics and socioeconomic characteristics. Among disease-related variables, a number of proxies for illness severity were measured using comorbidity indices. Disease-related variables included treatment complexity and pill burden.

The incorporation of demographic, disease, and treatment characteristics into an easy-to-use nomogram facilitates individualised prediction of non-adherence to T2DM treatment. We developed and validated a new predictive model based on eight easily available variables (sex, marital status, education level, employment, SMBG, types of daily use hypoglycaemic pills, and daily dosing frequency of hypoglycaemic agents (pills, insulin, or GLP-1)) to predict medication nonadherence in patients with T2DM. If future nonadherence can be predicted, a time window for clinical intervention and treatment can be then obtained. Internal validation of the cohort showed good differentiation and calibration capability. In particular, our high C-index in interval validation suggests that

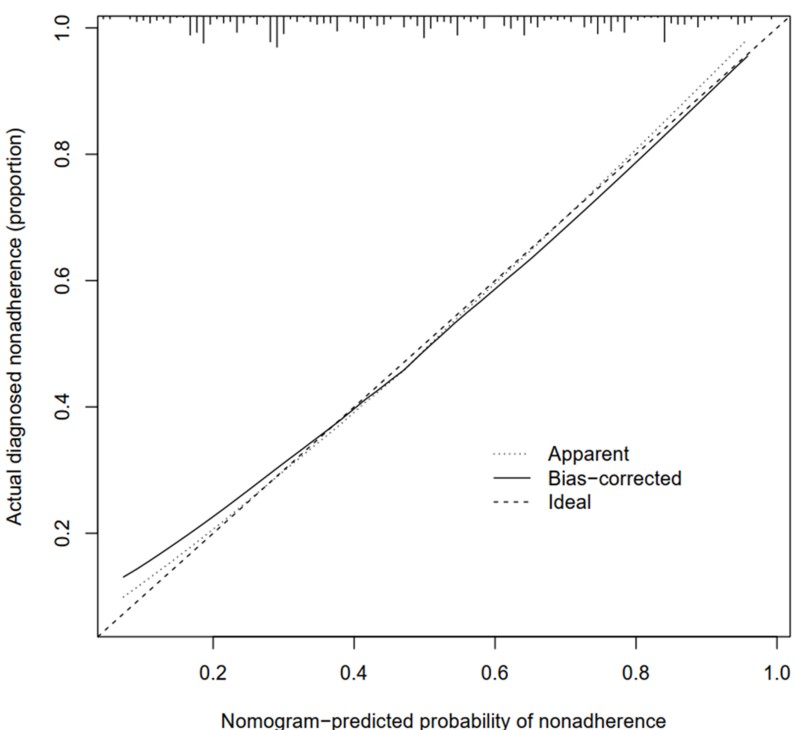

**Figure 3 Calibration curves of the nonadherence nomogram prediction in the cohort.** The *x*-axis represents the predicted medication nonadherence risk. The *y*-axis represents the actual diagnosed nonadherence. The diagonal dotted line represents a perfect prediction by an ideal model. The apparent line represents the entire cohort (*n* = 753). The solid line is bias-corrected by bootstrapping (B = 1,000 repetitions), represents the performance of the nomogram, of which a closer fit to the diagonal dotted line represents a better prediction.

the nomination table can be used extensively and accurately because of the fairly large sample size.

Hypoglycaemic medication nonadherence varies widely across individuals, and identification of sex differences for nonadherence may help healthcare providers customise effective interventions. Our investigation found that nonadherence to hypoglycaemic medication was higher in male patients (odds ratio: 0.54, 95% CI [0.34–0.85]).

Marital status strongly influenced adherence to medication. Several studies suggest that being married may positively influence adherence (*Cooper et al., 2005*; *Turner et al., 1995*; *Swett & Jerry, 1989*). However, results are severely limited by the very small number of the total who are single (74 *vs* 679). Previous research suggests that relative to those with high education levels, patients with low education levels regard the effects of medication as harmful and believe that they are overused (*Aflakseir, 2012*). Our data also show that having a lower education level may contribute to poorer adherence, probably because people with higher education can read and understand the importance of medication adherence.

The independent effect of employment status on treatment adherence has been elucidated in various disease settings. A prospective cohort study of patients who were treated with warfarin showed that employed patients were at increased risk of
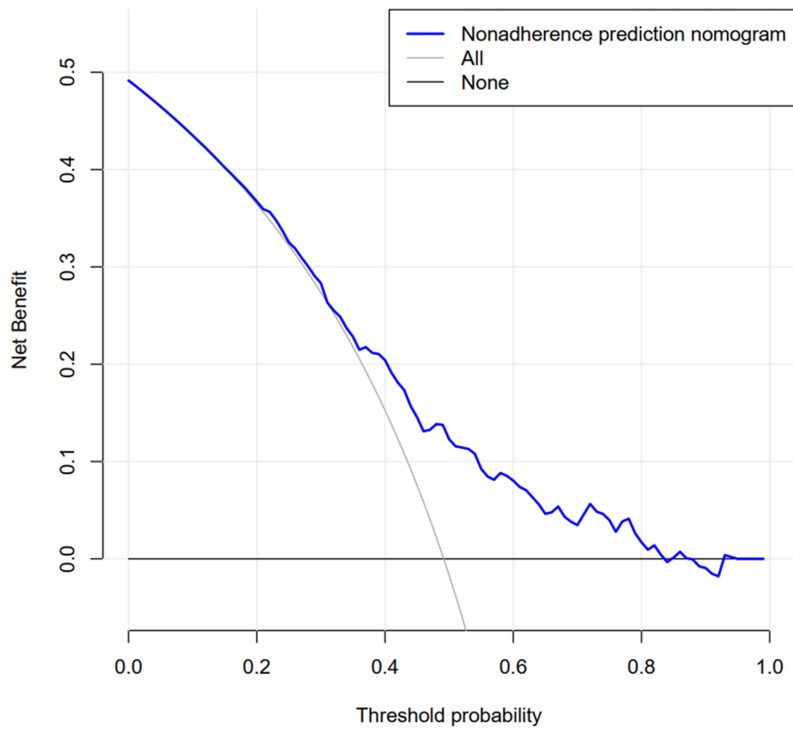

**Figure 4 Decision curve analysis for the nonadherence nomogram.** The x-axis measures the threshold probability, means that in the nonadherence nomogram, the probability of patient i medication non-adherence is recorded as Pi; When Pi reaches a certain threshold (recorded as Pt), it is defined as positive and intervention measures are taken; At this time, increasing the benefits of medication nonadherence patients (pros), while patients with adherence increase time and economic cost (cons). The y-axis is the net benefit minus the disadvantage, measures the net benefit. The blue line represents the medication nonadherence risk nomogram. The thin solid line represents the assumption that all patients are adherent to medication. The thick solid line represents the hypothesis that all patients are nonadherence. The area among the model curve "None line" and "ALL line", represents the clinical usefulness of the model.              

nonadherence to treatment relative than those who were unemployed and retired (*Platt et al., 2008*). Among patients with inflammatory bowel disease, men with lower medication adherence were more likely to have full-time employment (*Ediger et al., 2007*). However, our data show that employed patients with T2DM are at a greater risk of medication nonadherence than the unemployed. Work-related barriers, including being away from home, too busy, and irregular mealtimes, distracted and discouraged employed patients from compliance.

Our data show that adherence may be influenced by distance to the hospital, with longer distances causing less adherence. Patients are likely to fill and pick up prescriptions at pharmacies that are close to places of work or along commuting routes. Although a longer distance to travel to a hospital may appear as a barrier, access to transportation is more important than distance, and convenient transportation may negate the effects of increased distance (*Mooyoung, Suarez & Adamson, 2018*). Further work is needed to determine whether vehicles or distance have a greater impact on adherence.

Self-care practices, including self-monitoring of blood glucose (SMBG), are crucial in diabetes management. SMBG is recommended for monitoring glycaemic status (*Goldstein et al., 2004*). SMBG incorporated with structured brief counselling provided patients with a tool for taking on more self-control and improving medication adherence.

Surprisingly, we found positive correlations between disease duration and medication nonadherence. Previous studies on medication adherence in patients with chronic diseases have found that patients frequently stop taking medications because of ineffective or unpleasant side effects. Notably, in our study, the side effects that patients perceived as interfering with medication adherence tended to be chronic and were not generally reported by the patient to his or her primary care physician. In addition, diabetic symptoms are likely to disappear if the course is followed for a prolonged period. In asymptomatic conditions, patients might consider that they do not need medication and fail to fill their prescriptions (*Miller, 1997*).

In line with previous reports (*Weeda et al., 2016*), our study shows that twice- or thrice-daily dosing does not provide a significant advantage over once-daily dosing, with once-daily dosing correlating with better compliance. Thus, simplified dosing may improve patient adherence. Diabetes-related pill burden could be impacted by manufacturers by developing formulations once daily (or even less frequent) dosing. The dosing frequency should be considered when selecting a prescription. Of note, relative to once-daily dosing, multiple-daily dosing (≥4 times daily) is likely to be achieved with better compliance. We speculate that this may be due to the following reasons. T2DM risk onset increases during the middle years of life (*Arai et al., 2005*). Thus, the duration of diabetes increases with age. Meanwhile, the incidence of microangiopathy and related disorders potentiates medication adherence. However, further investigations are needed to verify this hypothesis. Accurate prognostic assessment will guide physicians when assessing patients' medication nonadherence and help them to take timely interventions, to prevent testing at low risk, and to avoid treatment delay or discontinuation when there is a high likelihood of a favourable net benefit. Since predicting nonadherence by individual patients is difficult, appropriate measurements and multifaceted interventions may be the most effective strategies against unsatisfactory adherence (*Kardas, Lewek & Matyjaszczyk, 2013*). More attention should also be paid to employed single men with a long disease duration (>1 year). Additional strategies may be needed to inform employed single men on the significance of using medications as prescribed. Approaches that link the use of diabetic drugs to routine daily activities, for example, after a meal, and going to bed should be applied to increase medication adherence. Other interventions include provision of plastic daily dosing containers and reminders to help patients adhere to prescribed medications, such as daily text or emails. Sufficient resources are needed to identify strategies to promote medication adherence and, hence, effective glycaemic control.

## Limitations

Some factors may limit the generalisability of our findings. First, the sample size may not be sufficiently large, which may underlie the null effect findings. Second, the study was conducted at a single location; a multicentre study may improve the generalisability of our

findings. Other elements, such as social support related to nonadherence, were not closely examined. Although the factors discussed above can improve medication adherence, longitudinally designed studies are advocated to reveal strategies with the best outcomes for this specific group of patients, and to identify the long-term outcomes of nonadherence to diabetes medication. Third, we did not externally validate the nomogram; hence, it cannot be applied to T2DM populations in other regions and countries. Fourth, there are potential drawbacks to using a nomogram. The exact answer to whether nomogram-assisting decisions improve patient prognosis lies in prospective assessment—randomly assigning patients to nomogram-based or non-nomogram-based decisions and comparing outcomes. However, the prospective validation of each nomogram form prior to use is impractical. Hence, despite its good performance, nomograms may lack clinical utility and assessments of whether the nomogram improves patient and physician satisfaction, quality of life, and outcomes are often ignored.

Despite these limitations, our model predicted patient medication nonadherence with a relatively high $C$-index, which when combined with personalised interventions can improve medication adherence and generate potentially significant cost savings by reducing the levels of nonadherence.

## CONCLUSION

This study revealed a high rate of anti-diabetes treatment nonadherence. Medication nonadherence was significantly associated with long disease duration, employed single man with less formal education, and far hospital distance. Strategies that may improve adherence include amending medicine availability, promoting once-daily dosing, and widespread medical counselling. This is only validated in a Chinese population at a single centre, as the study cannot yet support wider predictive value for the nomogram. Our nomogram requires external validation. Future research will determine if interventions applied on the basis of the nomogram will decrease medication nonadherence rates and promote treatment outcomes.

## ACKNOWLEDGEMENTS

The authors appreciate all the study participants who voluntarily participated in this study for their contribution to the success of this work.

### Funding

This work was supported by the National Key R & D Program of China (NO:2018YFC 1314000). The funders had no role in study design, data collection and analysis, decision to publish, or preparation of the manuscript.

### Grant Disclosures

The following grant information was disclosed by the authors:
National Key R & D Program of China: 2018YFC 1314000.

## Competing Interests

The authors declare that they have no competing interests.

## Author Contributions

- NaRen QiMuge conceived and designed the experiments, performed the experiments, analyzed the data, prepared figures and/or tables, authored or reviewed drafts of the paper, and approved the final draft.
- Xu Fang performed the experiments, analyzed the data, prepared figures and/or tables, and approved the final draft.
- Baocheng Chang conceived and designed the experiments, analyzed the data, authored or reviewed drafts of the paper, and approved the final draft.
- Dong Mei Li conceived and designed the experiments, analyzed the data, authored or reviewed drafts of the paper, and approved the final draft.
- Yuanyuan Li performed the experiments, prepared figures and/or tables, and approved the final draft.

## Human Ethics

The following information was supplied relating to ethical approvals (*i.e.*, approving body and any reference numbers):

Inner Mongolia People's Hospital ethical committee approved the study (202000406).

## Data Availability

The raw data are available in the Supplemental File.

## Supplemental Information

Supplemental information for this article can be found online at http://dx.doi.org/10.7717/peerj.13102#supplemental-information.

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
