# Peer review of "Predicting population: development and validation of a new predictive nomogram for evaluating medication nonadherence risk in a type 2 diabetes"

_PeerJ, doi:10.7717/peerj.13102_

## Round 0.1 · original submission · Major Revisions

As you will, both reviewers feel that the work has merit. Reviewer-1 has some straightforward suggestions, and reviewer-2 has identified a fair amount of required changes. These would appear to be fairly straightforward too, but the list is extensive and in particular I draw to your attention the discussion raised about the odds ratio's by the second reviewer, and the need to elaborate on what the nomogram would bring to the field.

It is essential that your address each and every point raised by. both reviewers and clearly provide an explanation of your changes in a rebuttal letter.

Thanks for sharing this work with us. I hope you will find these comments helpful.

Reviewer 1 ·

Basic reporting

This is a cross sectional study in a Chinese population to develop and validate a medication compliance nomogram for adherence to prescribed therapy in T2D.

Overall, it is well presented and written with clear communication of ideas. There are a few minor errors in the English within the text such as lack of spacing between some words. Care should be taken to proof read and correct this. Otherwise it is written in clear and comprehensible English. The paper is structured appropriately with supporting figures and tables. References are appropriate and used to support arguments/methodology

Experimental design

The authors aim to create and validate a predictive nomogram for use to predict medication compliance in people with T2D in a single centre Chinese population. They include 753 patients from a single centre in their analysis. There is a reasonable breadth of ages and a reasonable balance between sexes. The study was cross-sectional between July to December 2020. The authors use a validated tool for assessment of medication compliance (MMAS-8) and provide appropriate references to support this. Questionnaires were used in conjunction with this to collect basic demographic and medical information. This was then correlated with compliance scores to predict which - if any - features correlate with compliance in this population. Optimisation of features used for the nomogram was derived using LASSO. A prediction model was then created using the appropriate demographic and other features. This model was then assessed for performance using appropriate tools including c-index. This appears to confirm that the model the authors have created is predictive in this population. They then propose using a 12% threshold of likely non-compliance as a target to aim specific interventions. The authors provide suggestions around this but not any great detail.

Overall, the methodology is valid and well described. This would allow replication in other centres.

Validity of the findings

The findings presented are valid and well described and would allow the process to be replicated. There are a number of limitations - most of which the authors address in a succinct limitation section at the end of the manuscript. This is a relatively small cross-sectional study which of course brings limitations. Some of the conclusions - such as compliance being less in single people - are severely limited by the very small number of the total who are single (74 vs 679). The authors could make this limitation clearer. In their conclusion, I would recommend that the authors are more specific in stating that this is only validated in a Chinese population at a single centre as the study cannot yet support wider predictive value for the nomogram. The authors describe some interventions they might use to increase compliance and conclude that future research to determine the clinical validity of the nomogram in a (presumably) prospective study. I think the findings are reasonable, if not entirely novel. I would recommend the authors look at the literature in the area of compliance in multiple daily dosing of diabetes medications again as there is literature examining frequency of medication administration and compliance. I would recommend they remove the line stating that there is no other literature to support this contention, as this is not the case.

Additional comments

Some particular statements are speculative and questionable around the differences in compliance by sex. The authors note that men are perhaps more likely to be non-compliant as they have "busier professional and social lives". Could it also be the case that men are less organised than women? Could there be other health behaviours in men to explain this? This statement appears slightly clumsy at best; offensive to women at worst - particularly without a reference to support the author's speculation in their specific population. My suggestion would be to remove it.

·

Basic reporting

The paper is well written with good grammar and punctuation.

The introduction is quite limited and while it gives context there is not much detail. Nomograms are not mentioned in the introduction, despite being at the core of the paper. They need to be introduced and explained at an early stage, especially given the authors’ claim that this is the first time they have been used in the area of medication adherence of diabetes patients.

The raw data has been shared but is all in Chinese characters. For this to be published in an English language journal I think the data should be in English.

Table 1 has a rogue bracket in the first row.

Table 2 – two decimal places are sufficient for presenting odds ratios and CIs. Three decimal places are sufficient for p-values. I would argue that there is no need for the p-values since the CIs give all of the required information.

The figures are all relevant but are very poorly explained in the text. The results section refers to each figure and table but gives very little or no interpretation of any of them. This makes it difficult for the reader to understand what they show and how it corresponds to the conclusions made, particularly for readers without expert knowledge of the LASSO approach.

The legend for Figure 1B says there is a vertical line drawn at the value selected in the fivefold cross-validation, but there is no line.

Figure 2 needs a clear explanation in the legend and in the text. While this type of visualisation may be understandable to someone in the oncology field who routinely works with nomograms, a reader without this background will not understand what the figure is showing.

The legend for Figure 3 does not explain what the ‘Apparent’ or ‘Bias-corrected’ lines represent.

Figure 4 is not well explained at all. On line 98 the authors mention ‘threshold probabilities’ but do not explain what they are. The text does not make clear where the patient and doctor threshold probabilities come from that are mentioned on line 128. It is not clear what the ‘scheme’ mentioned on line 130 is. It is not clear what the ‘intervention all-patients scheme and the ‘intervention non-scheme’ are.

Lines 76-77 in the ‘Adherence assessment’ section say ‘We categorized adherence as follows: low adherence = <6, medium adherence = 6 - <8, and high adherence = 8.’ Should medium be 6-7 and high be >=8, since it appears that the maximum score can be 11 (7 binary questions plus one Likert scale with a maximum of 4)?

Line 78 in the ‘Adherence assessment’ section: please replace ‘moderate adherence’ with ‘medium adherence,’ assuming that this is corresponding to the ‘medium adherence’ category defined in the previous sentence.

Experimental design

The research question is clear and seems to fill a knowledge gap.

It is not clear why the authors have fitted a logistic regression subsequent to the LASSO model. The latter would be sufficient to provide a model with appropriate odds ratios and confidence intervals. They should clarify why they have done this.

It should be made clearer how the test and validation subsets of the data are used in the analysis and at what stages each are used.

Validity of the findings

The authors give no indication of the potential drawbacks of using a nomogram (Balachandran VP, Gonen M, Smith JJ, DeMatteo RP. Nomograms in oncology: more than meets the eye. Lancet Oncol. 2015;16(4):e173-e180. doi:10.1016/S1470-2045(14)71116-7). These should be discussed.

Most of the discussion focuses on the variables that are found to be associated with nonadherence. The manuscript does not make clear what a nomogram adds to the field beyond just carrying out a logistic regression and this needs to be added. Furthermore, the authors should give examples of how the nomogram might be used in practice for individual patients.

Despite the extensive discussion on which variables are associated with nonadherence, there is no mention of the magnitude of the associations, based on the odds ratios and confidence intervals. Some of the odds ratios are very imprecisely estimated (ie they have wide confidence intervals), but the text in neither the results nor discussion sections give any indication of this. The authors should give clear interpretation in the text and ensure that the reader is aware of the precision with which the odds ratios are estimated.

---

## Round 0.2 · Minor Revisions

Please see the comment from the reviewer regarding the use of an alternative text narrative. I am minded to agree. Please make this change and resubmit.

·

Basic reporting

Mainly well reported. Some issues with grammar and could do with a proof-read.

The rebuttal text to point 4 of my first review is better than the corresponding text included in the methods section of the manuscript and should replace it.

Experimental design

No comment.

Validity of the findings

No comment.

Additional comments

I am happy with the authors' responses to my original review.

---

## Round 0.3 · accepted · Accept

Final minor changes made, thanks for attending to these points.

Congratulations on a nice study.